# K-RAS Acts as a Critical Regulator of CD44 to Promote the Invasiveness and Stemness of GBM in Response to Ionizing Radiation

**DOI:** 10.3390/ijms222010923

**Published:** 2021-10-10

**Authors:** Yi Zhao, Jae-Hyeok Kang, Ki-Chun Yoo, Seok-Gu Kang, Hae-June Lee, Su-Jae Lee

**Affiliations:** 1Department of Life Science, Research Institute for Natural Sciences, Hanyang University, Seoul 04763, Korea; zhaoyi0924@gmail.com (Y.Z.); jaehyeok1121@gmail.com (J.-H.K.); vanity0706@gmail.com (K.-C.Y.); 2Department of Neurosurgery, Brain Tumor Center, Severance Hospital, Yonsei University College of Medicine, Seoul 03722, Korea; seokgu9@yuhs.ac; 3Division of Radiation Effect, Korea Institute of Radiological and Medical Sciences, Seoul 01812, Korea; hjl22@kirams.re.kr

**Keywords:** CD44, K-RAS, ionizing radiation (IR), glioblastoma (GBM)

## Abstract

Radiation therapy is a current standard-of-care treatment and is used widely for GBM patients. However, radiation therapy still remains a significant barrier to getting a successful outcome due to the therapeutic resistance and tumor recurrence. Understanding the underlying mechanisms of this resistance and recurrence would provide an efficient approach for improving the therapy for GBM treatment. Here, we identified a regulatory mechanism of CD44 which induces infiltration and mesenchymal shift of GBM. Ionizing radiation (IR)-induced K-RAS/ERK signaling activation elevates CD44 expression through downregulation of miR-202 and miR-185 expression. High expression of CD44 promotes SRC activation to induce cancer stemness and EMT features of GBM cells. In this study, we demonstrate that the K-RAS/ERK/CD44 axis is a key mechanism in regulating mesenchymal shift of GBM cells after irradiation. These findings suggest that blocking the K-RAS activation or CD44 expression could provide an efficient way for GBM treatment.

## 1. Introduction

Glioblastoma multiform (GBM) is a grade IV glioma which is the most common and aggressive type of primary brain tumor of the central nervous system and has a poor prognosis. The median survival of GBM patients is less than 1 year [1,2], mainly due to its high propensity for tumor recurrence and chemotherapy resistance. Moreover, the recurrence of GBM is inevitable, because of the unclear management and dependent case [1]. Multimodal therapy including surgery followed by radiotherapy and chemotherapy has been well-known as the treatment approach for GBM patients [3,4]. Despite advanced therapeutic techniques and intensive multimodal therapy, the survival rate of most GBM patients still remain poor [5]. The reasons GBM is incurable with multimodal therapy are that tumor cells cannot be removed perfectly with surgery and chemotherapy is not effective because the blood-brain barrier (BBB) exists in the brain. Radiation therapy among these therapies has been reported that plays a crucial role in GBM therapy but also has tumor malignancy through the acquisition of radio-resistance [6,7]. Therefore, identifying the effective key molecular mechanism in response to radiation in GBM is essential to improve the poor prognosis of GBM patients and the development of therapeutic approach for GBM treatment.

K-RAS (Kirsten rat sarcoma viral oncogene homolog) is one of the three RAS oncogene family proteins, which belongs to a group of small GTP-binding proteins that cycles between GDP and GTP-binding states through the stimulation of membrane protein receptors such as GPCR [8]. K-RAS regulates cellular events including cell growth, differentiation, cell cycle, tumor development, invasion/migration, and metastasis by acting as a bridge in the signaling network that transduces a variety of signals to a wide arrange of downstream signaling pathways [8,9]. Multiple previous studies showed that activated K-RAS can transform primary cells to the highly metastatic phenotypes. RAS signaling plays a significant role in the regulation of cell–cell adhesion, matrix metalloproteinases, and genes alterations that regulate cell polarity to distinguish characteristics of the metastatic phenotype [6,10]. Activation of K-RAS signaling promotes tumor progression through stimulating multiple downstream signaling pathways, including extracellular signal-regulated kinase (ERK) and phosphatidylinositol 3-kinase (PI3K)/AKT signaling pathways [11]. Thus, blocking the K-RAS activation would provide a strategy for malignant tumor treatment.

In our previous study, we demonstrated that radiation-induced CD44 expression enhanced phenotypic changes and infiltration of GBM through SRC signaling [12]. However, the molecular mechanisms by which regulates radiation-induced CD44 expression still remain unclear. Therefore, in this study, we aim to explore the complete molecular mechanism for upregulation of CD44 expression in irradiated-GBM cells. Here, we found that the K-RAS/ERK axis upregulates CD44 expression in response to radiation in GBM. Specifically, ionizing radiation-induced K-RAS/ERK activation downregulates the expression levels of miR-185 and miR-202 which can bind to the 3′ UTR of *CD44* mRNA. The loss of miR-185 and miR-202, in turn, enhances CD44 expression. Finally, the K-RAS/ERK/CD44 axis promotes infiltration and mesenchymal shift of GBM cells through SRC activation, resulting in tumor aggressiveness. Hence, inhibition of K-RAS activation or CD44 expression would provide an effective therapeutic strategy for GBM treatment.

## 2. Results

### 2.1. Ionizing Radiation-Induced CD44 Expression Promotes Mesenchymal Shift and Stemness of GBM

In our previous study, we demonstrated that CD44 acts as a receptor for Hyaluronic acid (HA), and its expression can be induced by ionizing radiation. High expression of CD44 augments SRC kinase activity then leads to the infiltration and mesenchymal shift in irradiated GBM cells [12]. Western blot analysis and immunofluorescence staining assay showed that the CD44 expression was significantly increased in U87MG and U251MG cells in response to radiation (Figure 1A,B). The consistent results were observed in U87MG-orthotopic xenograft mouse model (Figure 1C,D). Additionally, FACS analysis and sphere formation assay showed that the percentage number of CD44^+^/CD24^−^ cells was increased in U87MG cells after radiation treatment (Figure 1E,F). Next, we knocked down *CD44* expression using siRNA and treated with or without radiation in U87MG and U251MG cells. The invasion and migration assay showed that upregulated invasion and migration ability of GBM cells after irradiation were decreased by transfected with CD44 siRNA (Figure 1G). Similarly, the expression of mesenchymal subtype markers, such as FN, Vim, and YKL-40 (a marker for the mesenchymal subtype of glioblastoma [13], also known as Chitinase-3-like protein 1, CHI3L1), was induced by irradiation and suppressed by blocking CD44 expression (Figure 1H). In order to confirm our results in vivo, we transduced U87MG cells with shRNA targeting CD44 and then injected intracranially into athymic BALB/c nude mice ((sh-Ctrl (*n* = 5); sh-Ctrl+IR (*n* = 4); sh-CD44+IR (*n* = 5)). After 15 days, xenograft tumors were locally irradiated with 2.5 Gy three times (Figure 1I). As expected, increased invasiveness of tumor was observed in irradiated mice compared to control mice. Moreover, IHC results showed that radiation enhanced the expression of mesenchymal markers such as, FN, Vim, and YKL-40 (a marker for the mesenchymal subtype of glioblastoma) in the xenograft GBM. However, these effects were diminished in xenograft tumors formed by U87MG cells transduced with CD44 sh-RNA (Figure 1J,K). Taken together, these results indicated that that radiation-induced CD44 expression enhances mesenchymal shift and infiltration of GBM cells both in vitro and in vivo.

### 2.2. Upregulation of CD44 Expression Is Regulated by Radiation-Induced K-RAS Activation in GBM

To investigate the mechanism which can be upregulated in response to radiation, we analyzed the expression profiles of irradiated-GBM cells from GEO dataset (GSE56937) using gene set enrichment analysis (GSEA). Here, we listed the top 10 signaling pathways that upregulated in radiation-treated U87MG cells with 8 Gy or 16 Gy based on the rank of the normalization of the enrichment score (NES) and found that K-RAS signaling was ranked at first and fourth in 8 Gy and 16 Gy-treated cells, respectively (Figure 2A,B). As reported, the RAS family (N-RAS, H-RAS, K-RAS), which is known as a small GTPase, is involved in cell survival and growth and commonly plays an oncogenic functional role in human cancer progression [8,14]. Subsequently, we performed western blot analysis to measure the expression of K-RAS, H-RAS, and N-RAS, then found that only K-RAS expression was increased after radiation treatment in U87MG cells (Figure 2C). To confirm this finding, we performed the RAS activation assay, and the consistent result was observed in radiation-treated U87MG cells (Figure 2D). Accordingly, we investigated that whether the CD44 expression was regulated by K-RAS activation in GBM cells after radiation treatment. Immunofluorescence staining analysis showed that the radiation can induce K-RAS and CD44 expression in the identical GBM cells both in vitro and in vivo (Figure 2E,F). Moreover, qRT-PCR and western blot analysis showed that the K-RAS and CD44 expression were increased in the irradiated-U87MG cells, while when we knocked down the K-RAS expression with si-RNA in irradiated-U87MG cells, the CD44 expression was decreased compared to the control groups (Figure 2G,H); however, CD133 which is a marker of stemness was partially induced by IR, but not significantly decreased by knock-down of K-RAS expression in IR-treated GBM cells (Figure 2I). Furthermore, the expression levels of CD44 were increased with the overexpression of K-RAS in U87MG cells (Figure 2J,K). Subsequently, the rescue experiment was performed using the U87MG cells transfected with K-RAS-overexpression vector alone or co-transfected with CD44 siRNA to check the EMT signature genes expression. Western blot analysis showed that the expression levels of EMT markers were increased in K-RAS-overexpressing U87MG cells, while these increases were inhibited after blocked CD44 expression with si-RNA (Figure 2L). Overall, these results showed that ionizing radiation-induced CD44 expression is mediated by K-RAS activation in GBM.

### 2.3. K-RAS/ERK/CD44 Axis Promotes the Invasiveness and Cancer Stemness Features of Irradiated GBM Cells

Next, to explore the effects of K-RAS on the infiltration of irradiation treated GBM cells, we performed the invasion assay and showed that the number of invasive cells was increased with the treatment of ionizing radiation and decreased after blocking the K-RAS expression with si-RNA (Figure 3A). Similarly, the FACS analysis also showed that radiation treatment enhanced the percentage of CD44^+^/CD24^−^ cells number, while it decreased with silenced the K-RAS expression in both U87MG and U251MG cells (Figure 3B and Appendix A). Moreover, the sphere formation assay showed that the IR-induced increases of sphere numbers were suppressed by blocking the K-RAS expression in U87MG cells (Figure 3C). Additionally, the expression of mesenchymal markers was also inhibited by silencing the K-RAS expression using qRT-PCR and western blot analysis (Figure 3D,E). These findings showed that the K-RAS is the main factor in regulating the radiation-induced CD44 expression and invasiveness in GBM. As reported, various different signal transduction pathways can be activated by RAS activation. Among these pathways, the MAPK (Mitogen-activated proteins kinase) is the best-studied pathway, which has known as a RAS major effector pathway and plays a key role in survival, drug resistance, and metastasis of cancer [15,16,17]. Previous studies showed that the MAPK signaling pathway mediates the radioresistance and inhibition of this pathway, and enhances the IR induced-cell toxicity in both mammary and prostate tumor cells [18,19]. To investigate whether K-RAS/ERK axis was involved in this mechanism in irradiated-GBM cells, we treated the ERK inhibitor, U0126 (10 μM), for 24 h into irradiated-U87MG cells to perform the invasion and migration assay and FACS analysis and sphere formation assay. Finally, we found that the invasive cells, the percentage of CD44^+^/CD24^−^ cells, and the number of spheres as well as the expression of mesenchymal markers were augmented with radiation treatment and were inhibited after treatment with U0126 (Figure 3F–I and Appendix A). Similar results were observed in the parallel experiments in K-RAS overexpressing U87MG cells, U251MG cells, and patient-derived X01 GBM cells treated with U0126 (Figure 3J–M and Appendix A). ICC staining assay revealed that the expression of CDH2 and Vim were decreased after blocking CD44 expression in K-RAS-overexpressing U87MG cells, which further confirm that K-RAS/ERK/CD44 axis regulated the invasiveness and stemness of IR-treated GBM cells (Figure 3N). Taken together, these data indicate that activation K-RAS/ERK signaling axis in response to IR enhances infiltration and stemness of GBM cells by upregulating CD44 expression.

### 2.4. miR-185 and miR-202 Negatively Regulates CD44 Expression in GBM Cells

Next, we explored how the K-RAS/ERK axis upregulates *CD44* expression in IR-treated GBM cells. Several previous studies reported the K-RAS and ERK activation downregulated the expression of microRNAs in various cancers [20,21], and the global downregulation of microRNAs expression was observed in multiple human cancers [22]. Moreover, microRNAs were critical in regulating tumor progression through downregulating mRNA expression of target genes [23,24]. Accordingly, we screened the microRNAs which have the potential to bind to the 3′ UTR regions of *CD44* mRNA using the public microRNA database and identified four microRNAs including miR185, miR-202, miR-204, and miR-590 as candidates (Figure 4A). qRT-PCR analysis was performed to check the expression levels of these microRNAs in U87MG cells after treatment with radiation or overexpression of the K-RAS vector. Finally, we found that the expression level of miR-185 and miR-202 were decreased both in IR-treated and K-RAS-overexpressing U87MG cells (Figure 4B,C). Subsequently, the qRT-PCR analysis showed that the expression levels of miR-185 and miR-202 were decreased after IR treatment or K-RAS overexpression vector, and it can be rescued by treatment with U0126 (Figure 4D,E). These results support that K-RAS/ERK axis downregulates the expression of miR-185 and miR-202. To investigate whether miR-185 and miR-202 can directly bind to the 3′ UTR regions of *CD44* mRNA, we performed a luciferase reporter assay. HEK293T cells were transfected with pGL3UC luciferase vectors containing the wide-type (WT) or mutant 3′ UTR of *CD44* mRNA and treated with miR-185 or miR-202 mimetics. The luciferase activity of HEK293T cells transfected with the wild-type CD44 3′ UTR pGL3UC vector was dramatically decreased after treatment with miR-185 or miR-202 mimetics; however, the luciferase activity of HEK293T cells transfected with the mutant CD44 3′ UTR pGL3UC vector was not changed after treatment with miR-185 or miR-202 mimetics (Figure 4F). Next, we performed invasion assay and found that the invasive ability of GBM cells was increased after treatment with the inhibitor of miR-185 and miR-202, respectively, compared to the control group (Figure 4G). FACS analysis and sphere formation assay also showed that the percentage of CD44^+^/CD24^−^ cells number and the number of spheres were increased with these microRNA inhibitors treatment using U87MG and the patient-derived X01 GBM cells, respectively (Figure 4H,I and Appendix A). Additionally, the expression level of mesenchymal markers and CD44 also increased in U87MG cells and patient-derived X01 GBM cells treated with microRNAs inhibitor (Figure 4J and Appendix A). Reversely, the CD44 expression was dramatically decreased after treatment with miR-185 or miR-202 mimetics in irradiated U87MG cells. Similar results were observed in K-RAS-overexpressing U87MG cells (Figure 4K,L). The invasion/migration assay and FACS analysis also showed that the invasion and migration ability, cancer stemness features of the cells were increased in U87MG cells and U251MG cells after IR treatment or K-RAS overexpression, respectively. While these features were suppressed after transfected miR-185 or miR-202 mimetics into the irradiated, or K-RAS-overexpressing U87MG and U251MG cells (Figure 4M–O and Appendix A). Similarly, the expression levels of mesenchymal markers were increased in irradiated U87MG and U251MG cells, and these were suppressed after treatment with miR-185 or miR-202 mimetics (Figure 4P and Appendix A). The consistent results were observed when we transfected the miR-185 and miR-202 mimetics into K-RAS-overexpressing U87MG and U251MG cells, respectively (Figure 4Q,R and Appendix A). Taken together, our data indicated that K-RAS/ERK axis negatively regulates miR-185 and miR-202 expression, in turn increasing CD44 expression to promote the mesenchymal shift and stemness of GBM cells.

### 2.5. CD44 and K-RAS Expression Is Positively Correlated in GBM

To determine the clinical relevance, we reanalyzed the public TCGA GBM database and found that *CD44* was highly expressed in the mesenchymal type of GBM patients compared to other types of GBM, which further supported that the expression level of *CD44* was positively associated with the mesenchymal phenotype of GBM (Figure 5A). Moreover, a positive correlation between *CD44* and *K-RAS* mRNA expression was observed in the GBM patients dataset from GEO database (GSE7696 and GSE108474) (Figure 5B). Next, we performed a tissue microarray assay (TMA) and found that high expression of K-RAS correlated well with the CD44 expression in GBM patient tissues (Figure 5C,D). Moreover, for the pre- and post-concurrent chemoradiotherapy (CCRT) treatment of GBM patients, the primary GBM patient tissue (pre-CCRT) showed a low expression of K-RAS and CD44, while a high expression of K-RAS and CD44 were observed in the recurrent GBM patient tissue (after-CCRT) using IHC and ICC staining analysis (Figure 5E). Kaplan-Meier survival analysis showed that the patients with high expression levels of CD44 or K-RAS showed a shorter survival time than those with low expression levels of CD44 or K-RAS (GSE108474, GSE4271) (Figure 5F,G). Furthermore, the patients with low expression of both CD44 and K-RAS showed longer survival times than the patients with the high expression of these genes (Figure 5H). Taken together, these findings demonstrated that the expression levels of K-RAS are positively correlated with the expression levels of CD44 in GBM patients, and the high expression of K-RAS and CD44 showed a poor clinical prognosis.

## 3. Discussion

GBM is the most aggressive and infiltrative brain tumor and shares a poor prognosis [25]. The standard treatment for GBM is surgery, followed by chemo- and radiotherapy. Radiotherapy is a common therapy used to destroy additional tumor cells after surgery. However, the surgery and the radiotherapy are extremely difficult because of the location and the invasive nature of the tumors, and the radioresistance and recurrence [26]. Therefore, the radioresistance and recurrence of GBM is still a grim issue for the GBM treatment. In our study, we identified an underlying mechanism, which would provide an effective approach and strategy for GBM treatment.

RAS is one of the well-known proto-oncogenes. Furthermore, the K-RAS, as an isoform of RAS family, is intensively investigated by researchers to explore its functional role in regulating the progression of human cancers [27]. K-RAS signaling pathway plays a key role in connecting the activation between other cellular signaling pathways which were involved in cell proliferation, differentiation, and survival in various cancers [28,29,30]. Additionally, K-RAS activation can induce invasion, metastasis, and angiogenesis of cells. The ionizing radiation-induced K-RAS/c-RAF/p38 pathway activation enhances cell invasion and migration ability in cervical cancer [31]; moreover, K-RAS or IR induced the phosphorylation of Y-box binding protein 1 (YB-1) promotes the repair of DNA-DSB and postirradiation survival through PI3K/AKT and MAPK/ERK signaling pathways in breast cancer [32]. Here, we found that irradiation can induce K-RAS expression and activation which stimulates its downstream ERK signaling activation. K-RAS/ERK axis plays an important role in regulating the malignancy of GBM cells in response to IR treatment. Therefore, blocking K-RAS activation may provide an effective strategy for cancer therapy.

CD44, a cell surface adhesion receptor, expresses a high level in various cancers and is involved in multiple signaling pathways in regulating cancer progression [33]. Accumulative of previous studies showed that CD44 can regulate cancer cell migration/invasion, angiogenesis, proliferation, and metastasis [34,35,36]. Additionally, high expression of CD44 was observed in GBM and associated with cell invasion through extracellular matrix interactions [37]. In our previous study, we demonstrated that irradiation-induced CD44 expression in GBM cells along with cell phenotype change and the increase of infiltration through CD44/SRC axis. Here, we aimed to explore the underlying mechanism that regulates CD44 expression in GBM cells after irradiation treatment. Previously, it was reported that fractionated IR could induce GSCs with the stemness phenotype by P62-mediated autophagy through the Wnt/β-catenin axis [38,39]. In our research, based on our previous study [12], we found that irradiated GBM cells can upregulate CD44 expression to induce stemness features. Through screening the hallmarks of radiation-treated GBM cells, using the public GBM databases, we found that K-RAS signaling pathway was upregulated in irradiated GBM cells. Based on this finding and the related experiments, we identified that GBM cells upregulate CD44 expression through K-RAS/ERK signaling pathway in response to radiation. Moreover, in our study, although IR can induce both CD44 and CD133 expression in GBM cells, while the upregulation of CD133 expression is not mediated by K-RAS signaling, radiation-induced CD133 expression might be regulated by other signaling pathway and this part need to be studied further. Here, we explored how K-RAS/ERK axis upregulates CD44 expression in response to IR treatment. A previous study reported that the CD44 expression was downregulated by miR-34a in prostate cancer [40]. In our study, through screening the predicated micoRNAs from the public microRNA databases, we found a total of four microRNAs including miR-202-5p, miR-185-5p, miR-204, and miR-590, which can potentially bind to the 3′UTR regions of *CD44* mRNA. Overexpression of miR-202 or miR-185 mimetics suppresses CD44 expression in irradiated GBM cells. Inversely, treatment with the inhibitor of miR-202 or miR-185 induced the expression of CD44 in U87MG cells. Based on these findings, we concluded that IR-induced K-RAS activation downregulates the expression levels of miR-202 and miR-185 to induce CD44 expression in GBM cells.

Multiple previous studies showed that CD44 played a significant role in regulating cell invasion, migration, and cancer stemness in various tumors [37,41]. Moreover, CD44 expression was upregulated in many cancer cells and recognized as a cancer stem cell marker [42]. As reported, the cancer cells that undergo EMT process can acquire cancer stem-like properties and showed an increase in CD44 expression [43]. Moreover, cancer cells with a mesenchymal phenotype show increased invasiveness and resistance to chemotherapy [44]. Although previous studies showed that K-RAS and CD44 played significant roles in various cancer, there is a lack of study of K-RAS and CD44 functional role in radiation-treated GBM. Here, we identified that high expression of CD44 enhances EMT and cancer stemness of irradiated GBM cells, leading to tumor aggressiveness. Collectively, in this study, we reported that radiation-induced activation of the K-RAS/ERK axis promotes the expression of CD44 by downregulation of miR-185 and miR-202 expression in GBM cells. Moreover, high expression of CD44 promotes the acquisition of invasiveness and stemness of GBM in response to IR. Hence, targeting the K-RAS/ERK/CD44 axis would provide an effective way for GBM treatment (Figure 5I).

## 4. Materials and Methods

### 4.1. Cell Culture

U87MG and U251MG glioma cells were obtained from the Korean Cell Line Bank (Seoul, Korea). Patient-derived X01 GBM cells, which were established previously from human GBM biopsy, were provided by Dr. Akio Soeda [45]. All U87MG, U251MG, and patient-derived X01 cells were cultured in Dulbecco’s Modified Eagle Medium (Invitrogen, Carlsbad, CA, USA), supplemented with 10% fetal bovine serum (FBS), penicillin (100 U/mL), and streptomycin (100 μg/mL). They were then incubated at 37 °C with 5% CO_2_.

### 4.2. Chemical Reagents

U0126 were obtained from Calbiochem (San Diego, CA, USA). Working concentration was 10μM and treated for 24 h before harvest.

### 4.3. Transfection

Polyethylenimine (PEI) or Lipofectamine^TM^ 2000 (Invitrogen) were used for cells transfection with DNA vector or siRNAs according to the manufacturer’s instructions. Briefly, total 2 μg of DNA vector or 40 μM siRNA was used for cell transfection. Next, the complexes were transfected into the appropriate cells for 48 h, then harvested for the subsequent experiments. All siRNAs were purchased from Genolution Pharmaceuticals (Seoul, Korea). All experiments were repeated at least three times.

### 4.4. Western Blot Analysis

The lysis buffer [40 mM Tris-HCl (pH = 8.0), 120 mM NaCl, 0.1% Nonidet-P40] supplemented with protease inhibitors was used to lyse the transfected cells. SDS-PAGE was used to separate the proteins, and then transferred these proteins to a nitrocellulose membrane (Amersham, Arlington Heights, IL, USA). The membrane was probed with the indicated primary antibody and developed with a peroxidase-conjugated secondary antibody. Finally, the proteins were visualized by enhanced chemiluminescence (ECL) (Amersham, IL, USA) according to the manufacturer’s instructions.

### 4.5. RNA Preparation and qRT-PCR Analysis

The TRIzol reagent (Invitrogen, Carlsbad, CA, USA) was used for total RNA isolation. Real-time quantitative PCR was performed using the KAPA SYBR FAST qPCR kit from KAPA Biosystems (Wilmington, MA, USA), according to the manufacturer’ s instructions. All analyses were performed using a Rotor Gene Q (Qiagen, Hilden, Germany) PCR cycler. mRNA expression levels were normalized to the expression of beta-actin. Results were expressed as fold change calculated using the ΔΔC_t_ method relative to the control samples. All the experiments were repeated at least three times. The primer sequences were shown in Appendix A.

### 4.6. Flow Cytometric Analysis

For the flow cytometric analysis, a total of 10^6^ control cells and/or appropriate condition treated cells were collected with trypsin digestion, and then washed and resuspended with 1× PBS. Subsequently, the cells were stained with R-phycoerythrin (PE)-conjugated anti-CD44 monoclonal antibody and FITC-conjugated anti-CD24 antibody (Miltenyi Biotec, Inc., Bergisch Gladbach, Germany) at 4 °C for 30 min. The data was analyzed using CellQuest software (BD Biosciences). All the experiments were repeated three times.

### 4.7. Immunofluorescence

For immunofluorescence assay, the transfected cells were fixed with 4% paraformaldehyde and permeabilized with 0.1% Triton x-100 in phosphate-buffered saline (PBS). They were then blocked with the blocking buffer (5% goat serum and 2% BSA in 1× PBS) for at least 30 min at RT. Subsequently, the cells were incubated with appropriate primary antibody at 4 °C overnight. The next day, after a total of three times washing in phosphate-buffered saline with 0.05% Tween 20 (PBST), the stained proteins were set with Alexa Fluor 488- or 594-conjugated anti-rabbit or anti-mouse secondary antibodies (Molecular Probes, Seoul, Korea). Nuclei were counterstained with 4′, 6-diamidino-2-phenylindole (DAPI; Sigma, St. Louis, MO, USA). They were then pictured using an Olympus Ix71 fluorescence microscope (Olympus, Seoul, Korea).

### 4.8. Immunochemical Staining Analysis

For the immunochemical staining assay, the paraffin-embedded tissue sections were deparaffinized in xylene, rinsed with 100, 95, 80, and 70% ethanol, and then washed with the flowing water. Epitopes were washed with 20 mg/mL protein kinase K in PBS with 0.1% Triton-X100. Sections were stained with H&E or immunostained with primary antibody at 4 °C overnight. The next day, after washing with PBS, biotinylated goat anti-rabbit IgG or anti-mouse IgG antibody was used for the sections, and incubated at RT for 1 to 2 h. After PBS washing, the ABC reagent (Vector Laboratories, Burlingame, CA, USA) was applied to the sections and the 3,3′-diaminobenzidine (Vector Laboratories) was used for the Color reaction, subsequently, stained with hematoxylin, and cleared with 70, 80, 95, and 100% ethanol and xylene. Then they were mounted with the Canada balsam. Images were captured using a DP71 digital imaging system on an IX71 microscope (Olympus, Seoul, Korea).

### 4.9. Irradiation

GBM cells or mouse brains were exposed to γ-rays using a 137Cs γ-ray source (Atomic Energy of Canada Limited, Mississauga, Canada) at a dose rate of 3.81 Gy/min. The dose of IR for GBM cells treatment was 2 Gy × 3 for this study, which was chosen based on the previously published clinical guidelines [12,25,46,47].

### 4.10. Invasion and Migration Assay

For the invasion assay, the filter inserts (Corning, NY, USA) were incubated at 37 °C for at least 30 min with the coating of 10 mg/mL growth factor-reduced Matrigel (BD Biosciences, CA, USA). Then, the cells were seeded into the coated filter inserts with the serum free medium. The medium with 10% FBS was added into the outside wells and incubated at 37 °C for 48 h. After incubation, the cells in the inner surface of the chamber were removed with a cotton swab. The invasive cells were fixed and stained using Diff-Quick kit (Fisher Scientific, Pittsburgh, PA, USA) and photographed. For the migration assay, it is similar as the invasion assay, except for the matrigel coating. The invasive and migrated cells were assessed by counting the number of the cells in three microscopic fields per well. The extent of invasion and migration was expressed as the average number of cells per field. All experiments were performed three times.

### 4.11. Sphere Formation Assay

For the sphere formation assays, the transfected or treated cells were resuspended in standard stem cell medium (SCM), which consisted of the following: DMEM/F12 (Invitrogen; Thermo Fisher Scientific, Inc., Waltham, MA, USA) supplemented with 1× B27 (Invitrogen; Thermo Fisher Scientific, Inc., Waltham, MA, USA), 20 ng/mL human recombinant epidermal growth factor (Sigma-Aldrich; Merck KGaA, Germany), and 20 ng/mL basic fibroblast growth factor (Sigma-Aldrich; Merck KGaA, Germany). The spheres were cultured at 37 °C in humidified air containing 5% CO_2_. The sphere size was determined using Motic Images Plus 2.0 software (Motic, Hong Kong) in three randomly chosen visual fields at different time points for 2 weeks. Colonies were photographed and their diameter was measured.

### 4.12. Luciferase Reporter Assay

Luciferase reporter assays were performed using vectors encoding putative target sites in the 3′ untranslated region (UTR). HEK293T cells were seeded into 60 mm dishes after reaching approximately 50% confluence and then co-transfected with reporter plasmid (1 μg), pRL-CMV-Renilla (Promega, Madison, WI) plasmid (1 μg) and miRNA using Plus reagent and Lipofectamine (Invitrogen) for 48 h. Luciferase activity was measured using a dual-luciferase reporter assay system (Promega) following the manufacturer’s instructions and normalized to Renilla luciferase activity.

### 4.13. Animal Experiments

Male athymic nude mice (Central Lab Animal Inc., Seoul, Korea), aged 6 to 8 weeks, were used in this study. U87MG GBM cells were implanted into the right frontal lobe of nude mice using a guide-screw system within the skull, as described previously [48,49]. A total of 2 × 10^5^ U87MG GBM cells were injected into each mouse (sh-Ctrl (*n* = 5); sh-Ctrl+IR (*n* = 4); sh-CD44+IR (*n* = 4)) using a multiple micro-infusion syringe pump (Harvard Apparatus, Holliston, MA, USA) at a speed of 0.5 μL/min. Mice with a similar weight and age were randomized to the experimental groups.

### 4.14. Gene Set Enrichment Analysis (GSEA) Dataset and Kaplan-Meier Analysis

Previously published microarray data from GSE56937, GSE7696, GSE108474, GSE4271 and TCGA GBM databases were used for reanalysis. Gene signatures obtained by comparing gene sets from either the Molecular Signature Database (MSigDB) database or published gene signatures were analyzed using GSEA analysis. Kaplan-Meier survival analysis using the KM plot program (http://kmplot.com/analysis/, accessed on 14 July 2021), as previously described [50].

### 4.15. Statistical Analysis

All experiments were repeated at least three times, and the data are reported as mean ± standard deviation (SD, represented by error bars). Statistical significance was evaluated through one-way analysis of variance (ANOVA) for multiple comparisons. A two-tailed Student’s t-test was used to compare data between two groups in GraphPad Prism software 8.0. *p* values < 0.05 were considered significant. The association between CD44 and K-RAS expressions of GBM patients’ tissues were investigated using χ^2^ test. Survival analysis was performed using the Kaplan–Meier method to calculate the survival rates among different groups. These groups were then compared using the log-rank method.

## Figures and Tables

**Figure 1 ijms-22-10923-f001:**
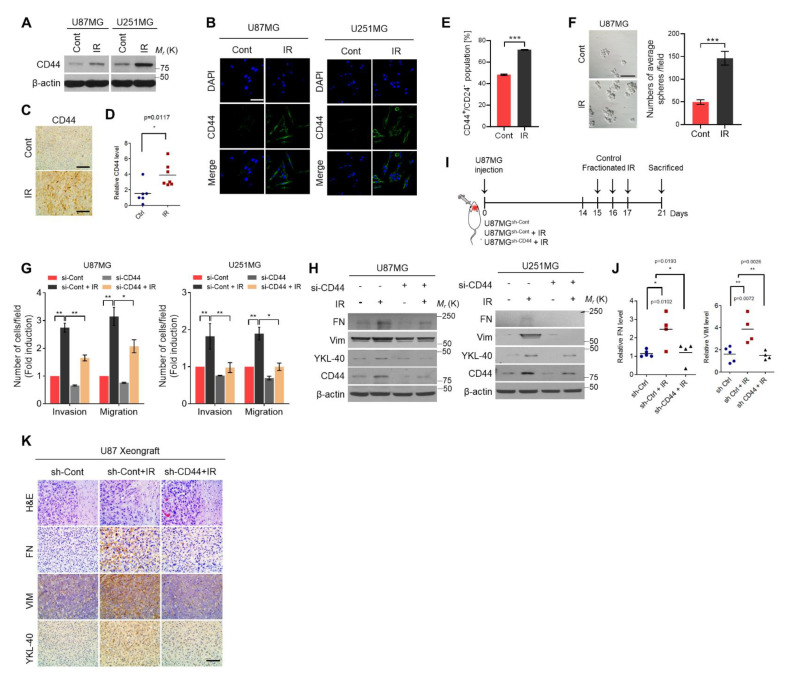
IR induced CD44 expression and promoted mesenchymal shift of GBM. (**A**,**B**). Western blot analysis and ICC staining assay were performed to check the CD44 expression in IR treated-U87MG cells. (**C**). Representative IHC staining images of CD44 expression of in mice tissues with or without ionizing radiation treatment. (**D**). The graph showed the intensity scores of CD44. (**E**). FACS analysis of the percentage number of CD44^+^/CD24^−^ in IR treated-U87MG cells. (**F**). Sphere formation assay showed that the cancer stemness features of GBM was increased after treated with the IR. (**G**,**H**). Invasion and migration assay, western blot analysis were performed to check the invasiveness ability of cells and mesenchymal markers expression in IR-induced U87MG and U251MG cells along with blocking CD44 expression. (**I**). Schematic of in vivo experiments. A total of 2 × 10^5^ U87MG GBM cells were injected into each mouse (sh-Ctrl (*n* = 5); sh-Ctrl+IR (*n* = 4); sh-CD44+IR (*n* = 4)) using a multiple micro-infusion syringe pump (Harvard Apparatus, Holliston, MA, USA) at a speed of 0.5 μL/min. (**J**,**K**). Representative H&E and IHC staining images of FN and VIM expression in mouse tissues. The graph showed the intensity score of FN and VIM expression. β-actin was used as a control for normalization of expression. * *p* < 0.05, ** *p* < 0.01, *** *p* < 0.001, n.s., not significant. A two-tailed Student’s *t*-test was used to compare data between two groups, and ANOVA was used for multiple comparisons.

**Figure 2 ijms-22-10923-f002:**
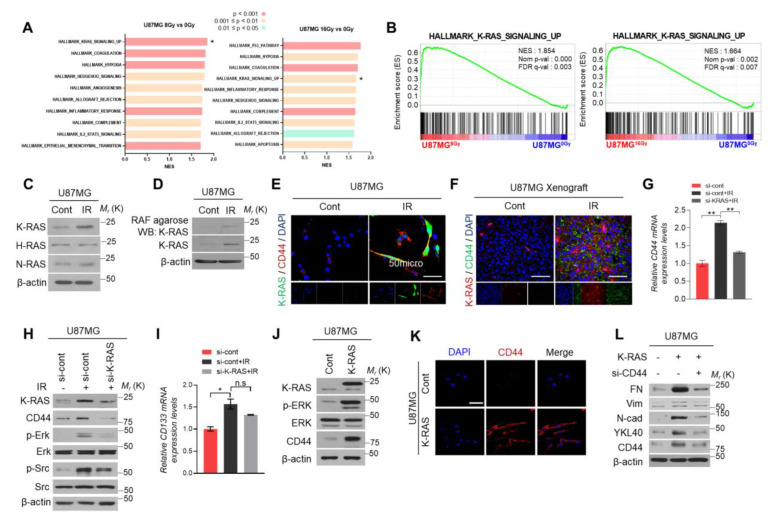
Radiation-induced K-RAS enhances CD44 expression in GBM. (**A**). GSEA analysis of hallmarks of irradiated-GBM cells using the data from GSE56937. (**B**). GSEA analysis indicated that irradiation treated U87MG cells positively correlated with K-RAS signaling pathway. (**C**). Western blot analysis of the expression of the members of RAS family. (**D**). Active K-RAS was measured by a Raf pull-down assay from irradiation treated or non-treated U87MG GBM cells. (**E**,**F**). Representative ICC staining images of K-RAS and CD44 expression in U87MG cells and mice tissues. (**G**,**H**). qRT-PCR and western blot analysis of CD44 expression using the ionizing radiation treated-U87MG cells with or without knocked down K-RAS expression. (**I**). qRT-PCR of CD133 expression using the ionizing radiation treated-U87MG cells with or without knocked down K-RAS expression. (**J**,**K**). Western blot analysis and ICC staining assay for checking the CD44 expression after overexpression of K-RAS in U87MG GBM cells. (**L**). Western blot analysis was performed to check the EMT signature genes expression in K-RAS-overexpressing U87MG cells with or without CD44 expression. β-actin was used as a control for normalization of expression. * *p* < 0.05, ** *p* < 0.01, not significant. A two-tailed Student’s t-test was used to compare data between two groups, and ANOVA was used for multiple comparisons.

**Figure 3 ijms-22-10923-f003:**
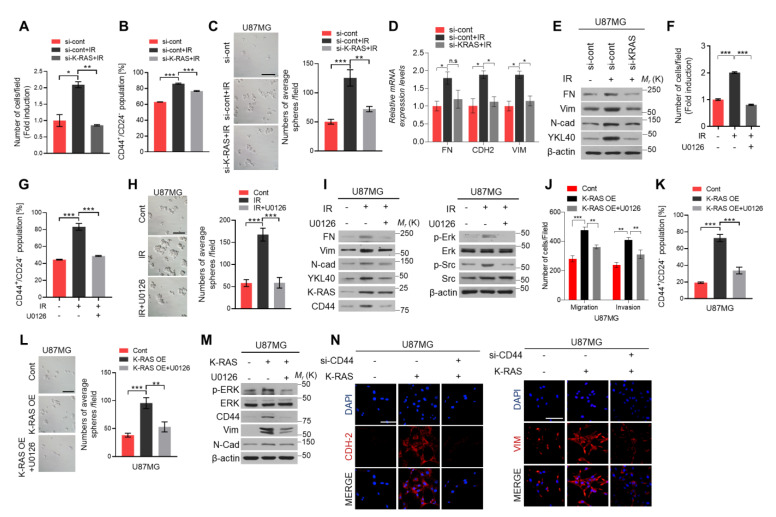
Radiation-induced acquisition of invasiveness and stemness is regulated by K-RAS/ERK/CD44 axis. (**A**). Invasion assay was performed using the radiation treated-U87MG cells with or without K-RAS expression. (**B**,**C**). FACS analysis and sphere formation assay were used to measure the percentage of CD44^+^/CD24^−^ cells number and the number of spheres using the same condition samples. (**D**,**E**). qRT-PCR and western blot analysis of EMT markers, FN, CDH2, and VIM, expression using the radiation treated cells with blocking K-RAS expression. (**F–H**). Invasion assay, FACS analysis, and sphere formation assay were performed to assess the invasiveness and stemness features with or without the ERK inhibitor treatment in IR treated-GBM cells. **I**. Western blot analysis was used to check the expression of mesenchymal markers, p-ERK and p-SRC. (**J–L**). Invasion assay, FACS analysis, and sphere formation assay were performed to assess the invasiveness and stemness features in the K-RAS-overexpressing U87MG cells with or without U0126 treatment, respectively. (**M**). Western blot analysis of p-ERK, ERK, CD44, Vim, and N-cad expression using the same condition samples of above. (**N**). Representative ICC staining images of CDH-2 and VIM expression after blocking CD44 expression in K-RAS-overexpressing U87MG cells. β-actin was used as a control for normalization of expression. * *p* < 0.05, ** *p* < 0.01, *** *p* < 0.001, n.s., not significant. A two-tailed Student’s t-test was used to compare data between two groups, and ANOVA was used for multiple comparisons.

**Figure 4 ijms-22-10923-f004:**
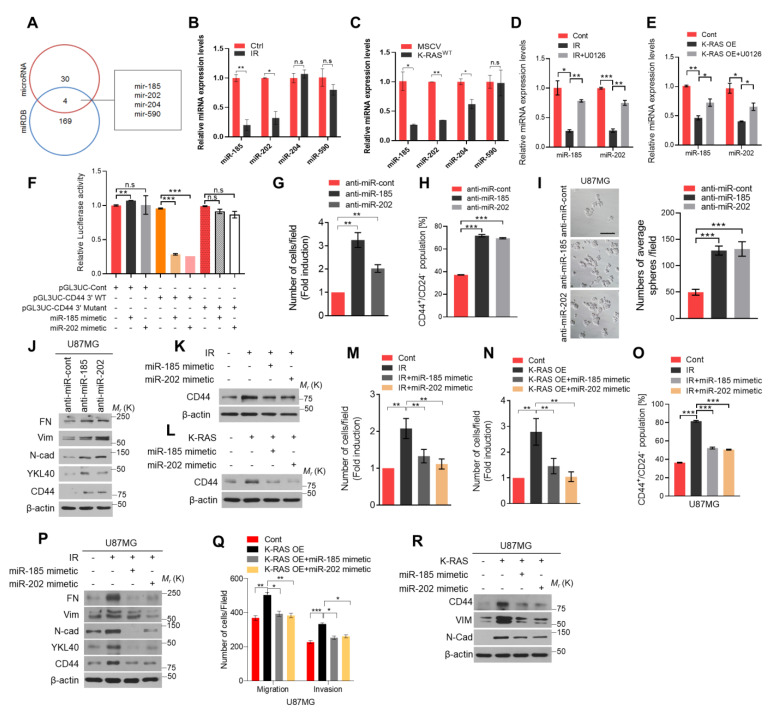
K-RAS-induced ERK activation regulates CD44 expression by suppression of miR-185 and miR-202 expression. (**A**). Schematic representation of miRNA prediction and screening using miRNA predictive websites (miRDB and microRNA). (**B**,**C**). qRT-PCR analysis of miR-185, miR-202, miR-204, and miR-590 expression using the IR-treated U87MG and K-RAS-overexpressing U87MG GBM cells, respectively. (**D**,**E**). qRT-PCR analysis of miR-185 and miR-202 expression using the IR-treated or K-RAS-overexpressing U87MG cells with or without U0126 (the working concentration was 10μM and treated for 24 h before harvest) treatment. (**F**). Relative luciferase activity after wild-type (WT) CD44-3’UTR and mutant CD44-3’UTR co-transfection with miR-185 or miR-202 mimetic in HEK293T cells. (**G–I**). Invasion assay, FACS analysis, and sphere formation assay were performed using the U87MG cells with the treatment of miR185 and miR-202 inhibitors at the control condition without IR treatment. (**J**). Western blot analysis of mesenchymal markers expression using the samples with the same experimental condition. (**K**,**L**). Western blot analysis of CD44 expression using the IR treated-U87MG cells and K-RAS-overexpression U87MG cells with or without miR185 and miR-202 mimetic transfection, respectively. (**M–P**). Invasion assay, FACS and western blot analysis were performed to check the mesenchymal shift and stemness features of IR-treated-U87MG cells with or without miR-185 and miR-202 mimetic transfection. (**Q**,**R**). Invasion and migration assay and western blot analysis were performed to measure the invasiveness ability of cells and the mesenchymal markers expression in U87MG cells with overexpression of the K-RAS alone or together with miR-185 and/or miR-202 mimetic. β-actin was used as a control for normalization of expression. * *p* < 0.05, ** *p* < 0.01, *** *p* < 0.001, n.s., not significant. A two-tailed Student’s t-test was used to compare data between two groups, and ANOVA was used for multiple comparisons.

**Figure 5 ijms-22-10923-f005:**
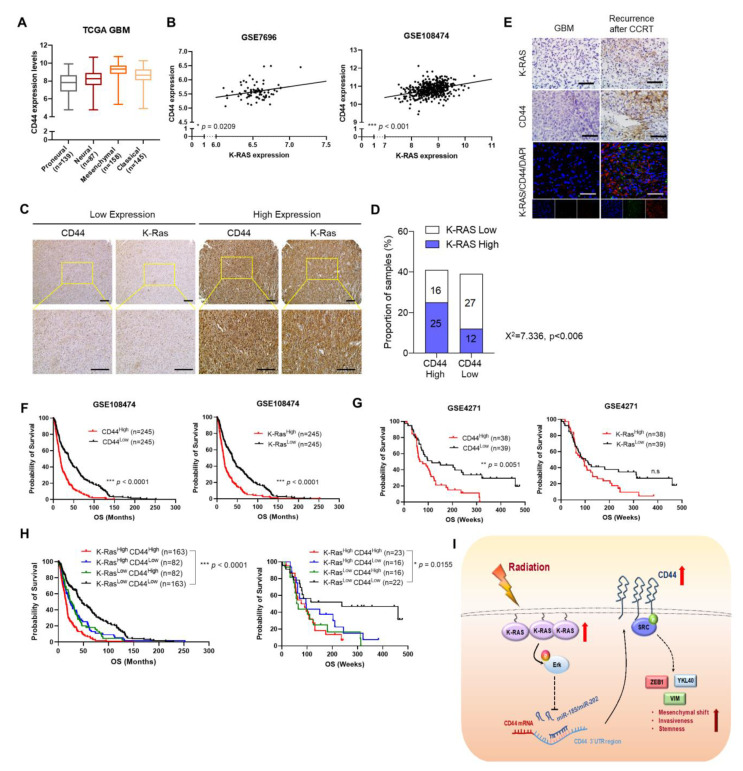
The clinical reference of the correlation between K-RAS and CD44 expression in GBM patients. (**A**). The CD44 expression in different subtypes of GBM patients using the data from TCGA GBM patient tissues database. (**B**). A positive correlation was observed between CD44 and K-RAS expression using the data from GSE7696 and GSE108474 databases. (**C**). Representative IHC staining images of CD44 and K-RAS expression in GBM patient tissues. (**D**). The association between CD44 and K-RAS expression in GBM tissues by χ^2^ test. (**E**). Representative IHC staining images of CD44 and K-RAS expression in the patient tissues from pre- and post-concurrent chemoradiotherapy (CCRT) treatment of GBM patients. (**F**,**G**). Kaplan-Meier survival analysis showed that high expression of CD44 and K-RAS showed a poor clinical outcomes. (**H**). The patients with low expression of K-RAS along with low expression of CD44 displays a longer survival time compared with the patients with high expression of these genes. (**I**). Schematic of the K-RAS/ERK/miR-185/miR-202/CD44 axis mechanism in GBM. The data from public GSE108474 and GSE4721 databases was used for reanalysis. * *p* < 0.05, ** *p* < 0.01, *** *p* < 0.001, n.s., not significant. A two-tailed Student’s t-test was used to compare data between two groups, and ANOVA was used for multiple comparisons. Survival analysis was performed using the Kaplan–Meier method to calculate the survival rates among different groups. These groups were then compared using log-rank method.

## Data Availability

All data needed to evaluate the conclusions in the paper are presented in the paper and/or the Appendix A. Additional data related to this paper may be requested from the corresponding author.

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
