# Peer review of "K-RAS Acts as a Critical Regulator of CD44 to Promote the Invasiveness and Stemness of GBM in Response to Ionizing Radiation"

_ijms, 2021, doi:10.3390/ijms222010923_

Round 1

Reviewer 1 Report

The paper presented by Zhao and colleagues show data regarding the involvement of K-RAS/MAPK in CD44 expression and invasiveness in glioblastoma cells, in response to radiation therapy. The data show are interesting and provide more insight on the molecular mechanisms of tumorigenesis of GBM cells post radiation.

However, many points are not clear and should be addressed to fully complete the picture the authors are presenting.

  1. Is not recommended use acronyms in the title, they use IR instead of radiotherapy.
  2. The data on Supplementary Figure 1 should be presented as main figure, since they are important data to show, even because in vivo data are shown.
  3. The data, suggest that CD44 is induced by radiation in GBM cells, which involves K-RAS/MAPK/ERK activation. Such effects are in fact hampered by the use of an ERK inhibitor, the well known U0126. Nevertheless, the model showed in figure 4I show an inhibition line (-----I) between ERK and the 2 miRNA under study. Still, no data on ERK modulation and miRNA 185 and 202 are provided to support such indication, in figure 3. Does ERK inhibition hamper K-RAS-induced miR185 mir-202 expression?Is an important point to assess, coupled to K-RAS overexpression that the authors already show.
  4. What is YKL40?Is not explained in the text even if this protein name appears in various figures.
  5. Data in Figure 2B do not show a clear reduction in CD44+/CD24- cells in siK-RAS+IR compared to the control. Please, show graph with independent assays.
  6. Is the assay in Figure 3D performed in control condition or post IR?It is not very clear.
  7. The Materials and methods provided are very poor and no much informations are provided. Which concentration of U0126 was used?For how much time?Which concentration of siRNA and miRNA mimetics have been used?For how long?How they choose the irradiation level?Based on what?All these informations are missing. Methods used are in general not very detailed.
  8. CD44 is a marker of stemness, but a very important one in Glioblastoma is CD133. Analysis of CD133 expression upon IR exposure, coupled to K-RAS modulation is strongly recommended to further provide evidences that IR induces stemness in GBM through K-RAS/MAPK involvement. Nevertheless, the effect of radiation on CD44 expression in GBM is already been published^: Life | Free Full-Text | Ionizing Radiation Induces Resistant Glioblastoma Stem-Like Cells by Promoting Autophagy via the Wnt/β-Catenin Pathway | HTML (mdpi.com); The authors should include these information and comment on that, in the discussion, highlighting the novelty that they bring with their manuscript.
  9. Authors indicate “mir mimics” in the figures, it is more appropriate “mimetic”
  10. Data on stemness are a bit poor, since only CD44 is shown to be affected. Apart from CD133 expression, mentioned above, the authors should analyse, for example, tumorspheres formation upon IR and K-RAs modulation, which is a well-established method to assess stemness in vitro.
  11. The statistic test indicated is T-Test. Nevertheless, it can be used to compare 2 different groups, but in many cases, the authors compared 3 or more groups, in which T-test cannot be used: as for example in Figure 3 I and J, M. Whenever more than 2 groups are compared, Anova test should be used. Please correct that and calculate again the statistics.
  12. Authors do not show a table with primer sequences for RT-qPCR. Please, provide that.
  13. English is very poor. Extensive editing is mandatory. There are statements not well written, with several same words repeated such as in lines 159-162. Other examples: line 90: “migrated ability”; line 60: “tumoregionesis”; Line 166: “we treated the ERK inhibitor”; and so on.

Author Response

Firstly, we would like to thank the editorial board for giving a chance to review our manuscript. We appreciate the reviewer’s insightful and critical assessment of our manuscript. Their comments have served as an important stimulus for making significant improvements. We appreciate the editorial and reviewer’s comments. We have revised the manuscript as per the reviewer’s suggestions. We believed these corrections will surely improve the quality of the manuscript and provide clarity to the readers.

Reviewer #1 (Comments to the Author): 

The paper presented by Zhao and colleagues show data regarding the involvement of K-RAS/MAPK in CD44 expression and invasiveness in glioblastoma cells, in response to radiation therapy. The data show are interesting and provide more insight on the molecular mechanisms of tumorigenesis of GBM cells post radiation.

However, many points are not clear and should be addressed to fully complete the picture the authors are presenting.

We appreciate the reviewer’s glowing interest regarding the importance of this manuscript.

  1. Is not recommended use acronyms in the title, they use IR instead of radiotherapy.

Response: We thank to the reviewer for this nice comment. As reviewer’s suggestion, we have revised the “IR” into “Ionizing Radiation” in our revised manuscript.

  1. The data on Supplementary Figure 1 should be presented as main figure, since they are important data to show, even because in vivo data are shown.

Response: We agree with the reviewer’s comment. We have revised the previously Supplementary Figure 1 as the main Figure 1 in our revised manuscript.

  1. The data, suggest that CD44 is induced by radiation in GBM cells, which involves K-RAS/MAPK/ERK activation. Such effects are in fact hampered by the use of an ERK inhibitor, the well known U0126. Nevertheless, the model showed in figure 4I show an inhibition line (-----I) between ERK and the 2 miRNA under study. Still, no data on ERK modulation and miRNA 185 and 202 are provided to support such indication, in figure 3. Does ERK inhibition hamper K-RAS-induced miR185 mir-202 expression?Is an important point to assess, coupled to K-RAS overexpression that the authors already show.

Response: We thank to the reviewer for this nice comment. In this study, we find that IR can induce K-RAS expression then activate its downstream MAPK/ERK signaling pathway, which inversely inhibit miRNA 185 and 202 expression. Loss of miRNA 185 and 202 expression then upregulates CD44 expression in IR-treated GBM cells. We agree with reviewer’s comment. As reviewer’s mention, we have performed the qRT-PCR analysis with the treatment of U0126 using the IR-treated or K-RAS-overexpressing U87MG cells to measure the miR-185 and 202 expressions. The data and description have shown in the revised Figure 4D and E and revised manuscript. 

  1. What is YKL40? Is not explained in the text even if this protein name appears in various figures.

Response: We thank to the reviewer for point it out. YKL-40, is a secreted glycoprotein, also known as Chitinase-3-like protein 1 (CHI3L1). It is expressed primarily in cells of mesenchymal origin and overexpressed in numerous aggressive carcinomas and sarcomas. Furthermore, the YKL-40 glycoprotein could be detected in the serum of patients with glioblastoma and other high-grade gliomas. Therefore, YKL-40 is a widely used as mesenchymal subtype marker for the mesenchymal subtype of glioblastoma (1-5). For this missing information has been added into our revised manuscript.

References:

  1. Joseph JV, Conroy S, Pavlov K, Sontakke P, Tomar T, Eggens-Meijer E, et al. Hypoxia enhances migration and invasion in glioblastoma by promoting a mesenchymal shift mediated by the HIF1alpha-ZEB1 axis. Cancer Lett (2015) 359(1):107–16.10.1016/j.canlet.2015.01.010

  1. Joseph JV, Conroy S, Tomar T, Eggens-Meijer E, Bhat K, Copray S, et al. TGF-beta is an inducer of ZEB1-dependent mesenchymal transdifferentiation in glioblastoma that is associated with tumor invasion. Cell Death Dis (2014) 5:e1443.10.1038/cddis.2014.395 

  1. Lu KV, Chang JP, Parachoniak CA, Pandika MM, Aghi MK, Meyronet D, et al. VEGF inhibits tumor cell invasion and mesenchymal transition through a MET/VEGFR2 complex. Cancer Cell (2012) 22(1):21–35.10.1016/j.ccr.2012.05.037

  1. Piao Y, Liang J, Holmes L, Henry V, Sulman E, de Groot JF. Acquired resistance to anti-VEGF therapy in glioblastoma is associated with a mesenchymal transition. Clin Cancer Res (2013) 19(16):4392–403.10.1158/1078-0432.CCR-12-1557

  1. Verhaak RG, Hoadley KA, Purdom E, Wang V, Qi Y, Wilkerson MD, et al. Integrated genomic analysis identifies clinically relevant subtypes of glioblastoma characterized by abnormalities in PDGFRA, IDH1, EGFR, and NF1. Cancer Cell (2010) 17:98–11010.1016/j.ccr.2009.12.020

  1. Data in Figure 2B do not show a clear reduction in CD44+/CD24- cells in siK-RAS+IR compared to the control. Please, show graph with independent assays.

Response: We thank to the reviewer for this comment. As reviewer’s suggestion, we have used the graph to present the CD44+/CD24- population percentage to clearly show the difference between each group. The data can be found in revised Figures such as, revised Figure 3B, G, and K, etc.

  1. Is the assay in Figure 3D performed in control condition or post IR?It is not very clear.

Response: We thank to the reviewer for this nice comment. The assay in previously Figure 3D were performed in control condition, moreover, we have added the information in our revised manuscript and figure legend.

  1. The Materials and methods provided are very poor and no much informations are provided. Which concentration of U0126 was used?For how much time?Which concentration of siRNA and miRNA mimetics have been used?For how long?How they choose the irradiation level?Based on what?All these informations are missing. Methods used are in general not very detailed.

Response: We thank to the reviewer for the nice comment. We have added the more detail information to the materials and methods section for readers easily understanding and repeat the experiments. The condition of U0126 is used at 10μM and treated for 24hr. The condition of siRNA and miRNA mimetic or inhibitors were 40 μM and transfected into the cells for 48hr with Lipofectimine 2000, before harvest.

For the radiation level used in this research, since previously clinical reports showed that the effects of IR were dependent on the total dose and fractionation ratio (1, 2). And other published clinical guidelines also showed that the 2 Gy per day for 5 days per week was used for the glioma cells (3). Additionally, combine these references and our previous study (4, Supplementary Figure 1A-D), which we found that the IR dose of 2 Gy × 3 or 2 Gy/d for 3d to GBM cells was enough to exhibit mesenchymal features with invasiveness. Thus, in this study, we used 2 Gy × 3 dose of IR for treating the GBM cells. As reviewer’s mention, we have added this missing information in our revised manuscript.

  1. Huber, S.M.; Butz, L.; Stegen, B.; Klumpp, D.; Braun, N.; Ruth, P.; Eckert, F. Ionizing radiation, ion transports, and radioresistance of cancer cells. Front. Physiol. 2013, 4, 212.
  2. Hubenak, J.R.; Zhang, Q.; Branch, C.D.; Kronowitz, S.J. Mechanisms of injury to normal tissue after radiotherapy: A review. Plast. Reconstr. Surg. 2014, 133, 49e–56e.
  3. Stupp, R.; Mason,W.P.; van den Bent, M.J.;Weller, M.; Fisher, B.; Taphoorn, M.J.; Belanger, K.; Brandes, A.A.; Marosi, C.; Bogdahn, U.; et al. Radiotherapy plus concomitant and adjuvant temozolomide for glioblastoma. N. Engl. J. Med. 2005, 352, 987–996.
  4. Yoo, K.C., et al., Proinvasive extracellular matrix remodeling in tumor microenvironment in response to radiation. Oncogene, 2018. 37(24): p. 3317-3328.

Reference 4: Figure S1A-D

  1. CD44 is a marker of stemness, but a very important one in Glioblastoma is CD133. Analysis of CD133 expression upon IR exposure, coupled to K-RAS modulation is strongly recommended to further provide evidences that IR induces stemness in GBM through K-RAS/MAPK involvement. Nevertheless, the effect of radiation on CD44 expression in GBM is already been published^: Life | Free Full-Text | Ionizing Radiation Induces Resistant Glioblastoma Stem-Like Cells by Promoting Autophagy via the Wnt/β-Catenin Pathway | HTML (mdpi.com); The authors should include these information and comment on that, in the discussion, highlighting the novelty that they bring with their manuscript.

Response: We thank to the reviewer for this constructive suggestion. As suggested by the reviewer, the effects of CD133 expression in response to IR and K-RAS expression were examined by qRT-PCR analysis. CD133 expression has partially increased by IR, but not significantly decreased by knock-down of K-RAS expression in IR-treated GBM cells. The data shown as below. Additionally, the sphere formation assay along with the FACS analysis were performed to further prove that the IR through K-RAS/ERK axis to upregulate CD44 expression then induce the stemness features of GBM. The results were shown in revised Figure 3G, H, K, L and Supplementary Figure 1, etc.

For the paper “Ionizing Radiation Induces Resistant Glioblastoma Stem-Like Cells by Promoting Autophagy via the Wnt/β-Catenin Pathway”, which is based on their previously finding that P62-mediated autophagy is associated with the Wnt/β-catenin pathway in glioma cells (1). Then they found that fractionated IR could induce GSCs with the stemness phenotype by P62-mediated autophagy through the Wnt/β-catenin axis. However, in our research, based on our previously study (2), we found that IR-treated GBM cells can upregulate CD44 expression to induce mesenchymal shift and stemness. Through screening the hallmarks of radiation-treated GBM cells, using the public GBM databases, we found that K-RAS signaling pathway was upregulated in IR-treated GBM cells. Base on this finding and the related experiments, we identified that GBM cells upregulate CD44 expression through K-RAS/ERK signaling pathway after the IR treatment. Moreover, in our study, although IR can induce both CD44 and CD133 expression in GBM cells, while the upregulation of CD133 expression independent on the K-RAS signaling, maybe other signaling pathways or the Wnt/β-Catenin pathway, reported by Tsai,C.-Y. et al. While this is not the main research in our study, and the further study is needed. For this part, we also added a description in our Discussion Section in our revised manuscript.

References:

  1. Chu, C.W.; Ko, H.J.; Chou, C.H.; Cheng, T.S.; Cheng, H.W.; Liang, Y.H.; Lai, Y.L.; Lin, C.Y.; Wang, C.; Loh, J.K.; et al. Thioridazine Enhances P62-Mediated Autophagy and Apoptosis Through Wnt/β-Catenin Signaling Pathway in Glioma Cells. Int. J. Mol. Sci. 2019, 20, 473.

  1. Yoo, K.C., et al., Proinvasive extracellular matrix remodeling in tumor microenvironment in response to radiation. Oncogene, 2018. 37(24): p. 3317-3328.

  1. Authors indicate “mir mimics” in the figures, it is more appropriate “mimetic”

Response: We thank to the reviewer’s comment. We have changed the “mimics” into “mimetic” in our revised figures and manuscript.

  1. Data on stemness are a bit poor, since only CD44 is shown to be affected. Apart from CD133 expression, mentioned above, the authors should analyse, for example, tumorspheres formation upon IR and K-RAs modulation, which is a well-established method to assess stemness in vitro.

Response: We thank to the reviewer for this nice comment. As reviewer’s suggestion, we have added the sphere formation assay upon IR and K-RAS modulation. The data can be found in our revised Figures and manuscript. Such as, revised Figure 3C, H, L, supplementary Figure 1F and G, etc.

  1. The statistic test indicated is T-Test. Nevertheless, it can be used to compare 2 different groups, but in many cases, the authors compared 3 or more groups, in which T-test cannot be used: as for example in Figure 3 I and J, M. Whenever more than 2 groups are compared, Anova test should be used. Please correct that and calculate again the statistics.

Response: We thank to the reviewer for pointing this out. In this study, statistical significance was evaluated through one-way analysis of variance (ANOVA) for multiple comparisons. A two-tailed Student’s t-test was used to compare data between two groups. P-values < 0.05 were considered significant. Additionally, for the data shown in previously Figure 3I, J, and M, we also analyzed by each indicated two groups. For example, the control groups (without IR and microRNA mimetic treatment) compared to the IR-treated group; or the IR-treated group compared to IR+miR-185 or 202 mimetic treatment group. The data shown in revised Figure 4M, N, and Q maybe easy for understanding. We have added the detail information about the statistical analysis in our revised manuscript.

  1. Authors do not show a table with primer sequences for RT-qPCR. Please, provide that.

Response: Thank to the reviewer for this nice comment. As reviewer’s suggestion, we have added the primer sequences information in our revised Supplementary Table S1.

  1. English is very poor. Extensive editing is mandatory. There are statements not well written, with several same words repeated such as in lines 159-162. Other examples: line 90: “migrated ability”; line 60: “tumoregionesis”; Line 166: “we treated the ERK inhibitor”; and so on.

Response: We thank to the reviewer for pointing this out. We are really sorry for this   mistake, we have corrected these issues in our revised manuscript.

Reviewer 2 Report

In this manuscript, the authors report that ionizing radiation (IR)-induced K-RAS/ERK activation elevates CD44 expression through downregulation of miR-202 and miR-185 expression. The authors performed a series of experiments to show that radiation accelerates stemness and EMT features of GBM through CD44 expression and confirmed the correlation between K-RAS and CD44 in the GBM patients. They also suggested that blocking the K-RAS activation or CD44 expression might be a therapeutic target to enhance the efficacy of radiotherapy for GBM. Overall, this study is quite good, including reasonable data and conclusions. However, there are also some concerns need to be addressed as outline below.

  1. For the Figure 2, it would be better to add U251MG cell line to perform the FACS analysis.

  1. The western blotting analysis should be performed with or without blocking CD44 expression in K-RAS-overexpressing U87MG cells to check the EMT markers expression.

  1. In Figure3K, vimentin staining is unclear and upregulation by IR is no apparent.

  1. The authors identified the upregulation of CD44 expression is mediated by suppression of microRNA. Is there any reason for selecting microRNA, not the direct upregulation by K-RAS/Erk pathway?

  1. The claim that IR increases the expression of CD44 is not apparent from the western blot data shown in Figure 3G and H.

  1. For the Figure 3L, it would be better to perform the same parallel experiments in U251MG cell line.

  1. In Figure 3, CD44 expression is regulated by miR-185 and miR-202; however, the additional experiments could be performed to confirm CD44 mRNA is direct target of miR-185 and miR-202.

  1. Most of the studies were performed using GBM cell lines including U87MG and U251MG cells. It would be helpful that verify the key data with human patient-derived primary GBM cells.

  1. For the Figure 4B, the statistics analysis is missing.

  1. IHC images in Figure4C should be replaced by clear images for identification.

  1. For the survival analysis data in Figure 4, the number of patients should be marked.

Author Response

Rebuttal Letter

Firstly, we would like to thank the editorial board for giving a chance to review our manuscript. We appreciate the reviewer’s insightful and critical assessment of our manuscript. Their comments have served as an important stimulus for making significant improvements. We appreciate the editorial and reviewer’s comments. We have revised the manuscript as per the reviewer’s suggestions. We believed these corrections will surely improve the quality of the manuscript and provide clarity to the readers.

Reviewer #2 (Comments to the Author): 

In this manuscript, the authors report that ionizing radiation (IR)-induced K-RAS/ERK activation elevates CD44 expression through downregulation of miR-202 and miR-185 expression. The authors performed a series of experiments to show that radiation accelerates stemness and EMT features of GBM through CD44 expression and confirmed the correlation between K-RAS and CD44 in the GBM patients. They also suggested that blocking the K-RAS activation or CD44 expression might be a therapeutic target to enhance the efficacy of radiotherapy for GBM. Overall, this study is quite good, including reasonable data and conclusions. However, there are also some concerns need to be addressed as outline below.

  1. For the Figure 2, it would be better to add U251MG cell line to perform the FACS analysis.

 Response: We thank to the reviewer for this nice comment. As reviewer’s mention, we have added another U251MG cell line and patient-derived X01 GBM cells to perform the FACS analysis. The data were shown in the revised Supplementary Figure 1A, B, C and E.

  1. The western blotting analysis should be performed with or without blocking CD44 expression in K-RAS-overexpressing U87MG cells to check the EMT markers expression.

Response: Thank to the reviewer for this nice comment. As reviewer’s suggestion, we have added these results in our revised Figure 2K, and also added the description in our revised manuscript.

  1. In Figure3K, vimentin staining is unclear and upregulation by IR is no apparent.

 Response: We thank to the reviewer for pointing it out. As reviewer’s suggestion, we have replaced the staining data of VIM for clearly show. The data can be found in revised Figure 3N.

  1. The authors identified the upregulation of CD44 expression is mediated by suppression of microRNA. Is there any reason for selecting microRNA, not the direct upregulation by K-RAS/Erk pathway?

 Response: We thank to the reviewer for this nice comment. Since previously studies showed that the K-RAS and ERK activation can downregulate microRNAs expression in cancers (1, 2). Furthermore, global downregulation of microRNA expression has been observed in many tumors (3). MicroRNAs were critical in regulating tumor progression through downregulating the target genes’ mRNA expression (4, 5). Based on these references, we hypothesized that whether K-RAS/ERK signaling pathway through microRNAs to regulate CD44 expression. The qRT-PCR analysis further supports our hypothesis that the K-RAS/ERK axis downregulates miR-185 and miR-202 expression in GBM. Additionally, the luciferase reporter assay strongly shows that the miR-185 and miR-202 can directly bind to the 3’UTR of CD44 promoter to downregulate its expression. The related data and description can be found in revised Figure 4B-F and manuscript. The references are shown as below:

  1. Sun HL, Cui R, Zhou J, et al. ERK Activation Globally Downregulates miRNAs through Phosphorylating Exportin-5.Cancer Cell. 2016;30(5):723-736. doi:10.1016/j.ccell.2016.10.001

  1. Jin, H., Jang, Y., Cheng, N. et al. Restoration of mutant K-Ras repressed miR-199b inhibits K-Ras mutant non-small cell lung cancer progression.J Exp Clin Cancer Res 38, 165 (2019). https://doi.org/10.1186/s13046-019-1170-7

  1. Lu, J., Getz, G., Miska, E. et al. MicroRNA expression profiles classify human cancers. Nature 435, 834–838 (2005). https://doi.org/10.1038/nature03702

  1. Fabian, M.R., N. Sonenberg, and W.J.A.r.o.b. Filipowicz, Regulation of mRNA translation and stability by microRNAs. 2010. 79: p. 351-379.
  2. Bartel, D.P.J.c., MicroRNAs: genomics, biogenesis, mechanism, and function. 2004. 116(2): p. 281-297.

  1. The claim that IR increases the expression of CD44 is not apparent from the western blot data shown in Figure 3G and H.

 Response: We thank to the reviewer for this nice comment. In order to support that IR can induce CD44 expression in GBM cells, we re-performed the western blotting analysis. And the data was shown in our revised Figure 4K, and L.

  1. For the Figure 3L, it would be better to perform the same parallel experiments in U251MG cell line.

 Response: Thanks to the reviewer’s comment. We have added the same parallel experiment using U251MG cells and showed the data in revised supplementary Figure 2E.

  1. In Figure 3, CD44 expression is regulated by miR-185 and miR-202; however, the additional experiments could be performed to confirm CD44 mRNA is direct target of miR-185 and miR-202.

 Response: We thank to the reviewer for this nice comment. We have performed the luciferase reporter assay to show that miR-185 and miR-202 can directly bind to the 3’UTR of CD44 promoter to regulate its expression. The data has shown in revised Figure 4F.

  1. Most of the studies were performed using GBM cell lines including U87MG and U251MG cells. It would be helpful that verify the key data with human patient-derived primary GBM cells.

Response: We thank to the reviewer for this comment. We used two kinds of GBM cell lines including U87MG and U251MG to demonstrate the regulatory mechanism of CD44 in response to radiation. As suggested by the reviewer, we confirmed the results using patient-derived X01 GBM cells. The additional data have shown in revised Supplementary Figure 1 and 2.

  1. For the Figure 4B, the statistics analysis is missing.

Response: Thank to the reviewer for this comment. We have added the statistics analysis in our revised Figure 5B.

  1. IHC images in Figure4C should be replaced by clear images for identification.

Response: Thank to the reviewer for pointing this out. We have replaced the IHC images in revised Figure 5C for clearly showing the data.

  1. For the survival analysis data in Figure 4, the number of patients should be marked.

Response: We thank to the reviewer for this nice comment. As reviewer’s suggestion, we have added the patients number in our revised Figure 5F, G, and H.

Round 2

Reviewer 1 Report

The reviewer really appreciates that the authors provide an extensively revised version of their manuscript, enriched with novel data, which confirm and reinforce their previous finding (e.g. tumorsphere formation assay).

Nevertheless, regarding the points that I have raised previously, I want to stress out on a few of them;

  1. U0126 concentration/time of exposure should be indicated in the Material and Methods section, not in the FIgure legend of Figure 3;
  2. The revised manuscript show data on cells derived from a patient, X01, which were not included in the previous version.  No information is reported on the cell culture condition of this primary cell sample, neither the characteristic of the patient/tissue nor if declared consent has been given for the use of the biological material;
  3. The data on CD133 are only mentioned in the discussion (lines 373-374): "Moreover, in our study, although IR can induce both CD44 and CD133 expression in GBM cells, while the upregulation of CD133 expression independent on the K-RAS signaling, maybe other signaling pathways or the Wnt/β-Catenin pathway were also involved in"; while RT-qPCR data are provided in the rebuttal letter. Please include these data in the manuscript or in supplementary information and include CD133 primer sequences in the provided tables. 
  4. Line 237: QRT-PCR: qRT-PCR

I regret to say that the English quality is still very poor. Expert scientific english editing service is strongly recommended. It is a real pity since the paper is of very high quality; better English would certainly improve the reading of the paper.

Ex.:

Next, we performed invasion assay, and found that the invasive ability of cells was increased when treated the inhibitor of miR-185 and miR-202, respectively, compared to the control group (Figure 4G). (line 250);

Moreover, in our study, although IR can induce both CD44 and CD133 expression in GBM cells, while the upregulation of CD133 expression independent on the K-RAS signaling, maybe other signaling pathways or the Wnt/β-Catenin pathway were also involved in"

Beginning section 2.4: To explore how the K-RAS/ERK axis upregulates CD44 expression in IR-treated GBM cells. As reported the K-RAS and ERK activation downregulated the expression of microRNAs in various cancers [22, 23]. And the global downregulation of microRNAs expression have been observed in human tumors.

Author Response

Firstly, we would like to thank the editorial board for giving a chance to review our manuscript. We appreciate the reviewer’s insightful and critical assessment of our manuscript. Their comments have served as an important stimulus for making significant improvements. We appreciate the editorial and reviewer’s comments. We have revised the manuscript as per the reviewer’s suggestions. We believed these corrections will surely improve the quality of the manuscript and provide clarity to the readers.

Reviewer #1 (Comments to the Author): 

Comments and Suggestions for Authors

The reviewer really appreciates that the authors provide an extensively revised version of their manuscript, enriched with novel data, which confirm and reinforce their previous finding (e.g. tumorsphere formation assay). Nevertheless, regarding the points that I have raised previously, I want to stress out on a few of them;

  1. U0126 concentration/time of exposure should be indicated in the Material and Methods section, not in the FIgure legend of Figure 3;

Response: We thank to the reviewer for this nice comment. As reviewer’s suggestion, we have indicated the treatment condition in the Material and Methods section.

  1. The revised manuscript show data on cells derived from a patient, X01, which were not included in the previous version. No information is reported on the cell culture condition of this primary cell sample, neither the characteristic of the patient/tissue nor if declared consent has been given for the use of the biological material;

Response: We thank to the reviewer for this constructive comment. Patient-derived X01 GBM cells which were established previously from human GBM biopsy were provided by Dr. Akio Soeda [1]. Patient-derived X01 cells were cultured in Dulbecco’s Modified Eagle Medium (Invitro-gen, Carlsbad, CA, USA), supplemented with 10% fetal bovine serum (FBS), penicillin (100 U/mL), and streptomycin (100 μg/mL). Then incubated at 37°C with 5% CO2. We have described the cell information and culture condition in Material and Methods section.

  1. The data on CD133 are only mentioned in the discussion (lines 373-374): "Moreover, in our study, although IR can induce both CD44 and CD133 expression in GBM cells, while the upregulation of CD133 expression independent on the K-RAS signaling, maybe other signaling pathways or the Wnt/β-Catenin pathway were also involved in"; while RT-qPCR data are provided in the rebuttal letter. Please include these data in the manuscript or in supplementary information and include CD133 primer sequences in the provided tables.

Response: We thank to the reviewer’s comment. As reviewer’s suggestion, we added these data in revised Figure 2I and added the primer sequences information in revised Supplementary Table S1.

  1. Line 237: QRT-PCR: qRT-PCR

Response: We thank to the reviewer for this correction. We have corrected this mistake.

  1. I regret to say that the English quality is still very poor. Expert scientific english editing service is strongly recommended. It is a real pity since the paper is of very high quality; better English would certainly improve the reading of the paper.

Response: We thank to the reviewer for this nice comment and are sorry for our English quality. We have proofread this manuscript and edited the grammar error.

We would like to sincerely thank this reviewer for his/her insightful suggestions and comments on our manuscript. We hope this revised version will satisfy his/her concerns. For clarity purposes, the revised text and new figure panels are marked up using the “Track Changes” function.

References

  1. Soeda, A.; Park, M.; Lee, D.; Mintz, A.; Androutsellis-Theotokis, A.; McKay, R. D.; Engh, J.; Iwama, T.; Kunisada, T.; Kassam, A. B.; Pollack, I. F.; Park, D. M., Hypoxia promotes expansion of the CD133-positive glioma stem cells through activation of HIF-1alpha. Oncogene 2009, 28, (45), 3949-59.